# Prognostic Role of Soluble and Extracellular Vesicle-Associated PD-L1, B7-H3 and B7-H4 in Non-Small Cell Lung Cancer Patients Treated with Immune Checkpoint Inhibitors

**DOI:** 10.3390/cells12060832

**Published:** 2023-03-08

**Authors:** Carlo Genova, Roberta Tasso, Alessandra Rosa, Giovanni Rossi, Daniele Reverberi, Vincenzo Fontana, Silvia Marconi, Michela Croce, Maria Giovanna Dal Bello, Chiara Dellepiane, Marco Tagliamento, Maria Chiara Ciferri, Lodovica Zullo, Alessandro Fedeli, Angela Alama, Katia Cortese, Chiara Gentili, Eugenia Cella, Giorgia Anselmi, Marco Mora, Giulia Barletta, Erika Rijavec, Francesco Grossi, Paolo Pronzato, Simona Coco

**Affiliations:** 1UOC Clinica di Oncologia Medica, IRCCS Ospedale Policlinico San Martino, 16132 Genova, Italy; carlo.genova@hsanmartino.it; 2Dipartimento di Medicina Interna e Specialità Mediche (DiMI), Università degli Studi di Genova, 16132 Genova, Italy; marco.tagliamento@edu.unige.it; 3Dipartimento di Medicina Sperimentale (DIMES), Università degli Studi di Genova, 16132 Genova, Italycortesek@unige.it (K.C.); chiara.gentili@unige.it (C.G.); 4UO Oncologia Cellulare, IRCCS Ospedale Policlinico San Martino, 16132 Genova, Italy; 5U.O. Epidemiologia Clinica, IRCCS Ospedale Policlinico San Martino, 16132 Genova, Italy; alessandra.rosa@hsanmartino.it (A.R.); vincenzo.fontana@hsanmartino.it (V.F.); 6U.O.C. Oncologia Medica 2, IRCCS Ospedale Policlinico San Martino, 16132 Genova, Italy; giovanni.rossi@hsanmartino.it (G.R.); chiara.dellepiane@hsanmartino.it (C.D.); lodozullo@gmail.com (L.Z.); eugenia.cella@hotmail.it (E.C.); giulia.barletta@yahoo.it (G.B.); paolo.pronzato@hsanmartino.it (P.P.); 7Dipartimento di Medicina, Chirurgia e Scienze Sperimentali Università di Sassari, 07100 Sassari, Italy; 8U.O. Patologia Molecolare, IRCCS Ospedale Policlinico San Martino, 16132 Genoa, Italy; daniele.reverberi@hsanmartino.it; 9U.O.S. Tumori Polmonari, IRCCS Ospedale Policlinico San Martino, 16132 Genova, Italy; silvia.marconi@hsanmartino.it (S.M.);; 10U.O.C. Bioterapie, IRCCS Ospedale Policlinico San Martino, 16132 Genova, Italy; michela.croce@hsanmartino.it; 11S.S.D. Animal Facility, IRCSS Ospedale Policlinico San Martino, 16132 Genoa, Italy; mariagiovanna.dalbello@hsanmartino.it; 12Dipartimento di Ingegneria Navale, Elettrica, Elettronica e delle Telecomunicazioni (DITEN), Università degli Studi di Genova, 16145 Genova, Italy; alessandro.fedeli@unige.it; 13U.O. Anatomia Patologica Ospedaliera, IRCCS Ospedale Policlinico San Martino, 16132 Genova, Italy; giorgia.anselmi@hsanmartino.it (G.A.); marco.mora@hsanmartino.it (M.M.); 14Unità di Oncologia Medica, Dipartimento della Medicina e Chirurgia, Università dell’Insubria, ASST dei Sette Laghi, 21100 Varese, Italy; erika.rijavec@asst-settelaghi.it (E.R.); francesco.grossi@uninsubria.it (F.G.)

**Keywords:** immune checkpoint inhibitors, NSCLC, PD-L1, B7-H3, B7-H4, soluble protein, extracellular vesicle, prognosis, platelets

## Abstract

The treatment of non-small cell lung cancer (NSCLC) has changed dramatically with the advent of immune checkpoint inhibitors (ICIs). Despite encouraging results, their efficacy remains limited to a subgroup of patients. Circulating immune checkpoints in soluble (s) form and associated with extracellular vesicles (EVs) represent promising markers, especially in ICI-based therapeutic settings. We evaluated the prognostic role of PD-L1 and of two B7 family members (B7-H3, B7-H4), both soluble and EV-associated, in a cohort of advanced NSCLC patients treated with first- (*n* = 56) or second-line (*n* = 126) ICIs. In treatment-naïve patients, high baseline concentrations of sPD-L1 (>24.2 pg/mL) were linked to worse survival, whereas high levels of sB7-H3 (>0.5 ng/mL) and sB7-H4 (>63.9 pg/mL) were associated with better outcomes. EV characterization confirmed the presence of EVs positive for PD-L1 and B7-H3, while only a small portion of EVs expressed B7-H4. The comparison between biomarker levels at the baseline and in the first radiological assessment under ICI-based treatment showed a significant decrease in EV-PD-L1 and an increase in EV-B7H3 in patients in the disease response to ICIs. Our study shows that sPD-L1, sB7-H3 and sB7-H4 levels are emerging prognostic markers in patients with advanced NSCLC treated with ICIs and suggests potential EV involvement in the disease response to ICIs.

## 1. Introduction

The therapeutic approach for patients with advanced non-small cell lung cancer (NSCLC) without targetable oncogenic drivers has been dramatically improved by the introduction of immunotherapy, both in a pre-treated and treatment-naïve setting. In particular, immune checkpoint inhibitors (ICIs) such as monoclonal antibodies targeting the programmed death 1/programmed death-ligand 1 (PD-1/PD-L1) pathway have demonstrated the ability to increase survival over chemotherapy in subgroups of patients, becoming the favored therapy after progression to platinum-based chemotherapy (nivolumab, pembrolizumab, atezolizumab) [1,2,3,4] and the treatment of choice as first line (pembrolizumab, atezolizumab) [5,6,7,8]. Although ICI-containing regimens show prolonged progression-free survival (PFS) and overall survival (OS), the discovery of new robust predictive factors of efficacy to guide treatment decisions is still an unmet need. To date, the expression of PD-L1 on tumor biopsy by immunohistochemistry is the only approved predictive biomarker, although it presents some drawbacks; its expression can vary spatially and over time and can be dependent on the pathologist’s interpretation [9]. Besides PD-L1, other immune checkpoint molecules, such as B7 homolog 3 (B7-H3) and B7 homolog 4 (B7-H4), are involved in the crosstalk during the immune response to cancer and, albeit playing a controversial role, have been reported to have prognostic value in solid tumors including NSCLC [10,11]. In a previous study conducted by our group, we found that the tumor expression of B7-H4 was associated with poor PFS and OS in NSCLC patients receiving nivolumab but not in those treated with chemotherapy [12]. All these molecules can also be found in soluble (s) form in the blood of patients with NSCLC, and their role is currently being investigated. In particular, high levels of these markers have been described as unfavorable factors in NSCLC [13], but to date there are no data on patients treated with ICIs. In addition, plasmatic sPD-L1 levels, as opposed to tumor tissue expression, have been linked to the reduced clinical benefits of nivolumab therapy [14]. In this context, emerging evidence suggests that the PD-L1 associated with circulating extracellular vesicles (EVs) plays a relevant role in immunosuppression [15,16]. EVs, acting as intercellular messengers by transferring protein and genetic materials, play an active role in tumor-associated immune cell communication [15], and in the immune response to ICIs [17,18]. To date, the involvement of EV-associated B7-H3 and B7-H4 in first- and second-line-treated NSCLC patients is still unknown. The present study aims to: (i) explore the role of sPD-L1, sB7-H3 and sB7-H4 as prognostic markers in patients with advanced NSCLC treated with nivolumab or pembrolizumab; (ii) assess the involvement of EVs expressing PD-L1, B7-H3 and B7-H4 in the mechanisms of the response to ICIs (Figure 1).

## 2. Materials and Methods

### 2.1. Patient Enrolment and Sample Collection

This exploratory study was based on an analysis of 182 consecutive patients with advanced NSCLC receiving ICIs from May 2015 to May 2019 in a mono-institutional translational research project approved by the local ethics committee (registry number: P.R. 191REG2015). Written informed consent was obtained from each patient. The patients were treated with pembrolizumab (Pembro cohort, PC) 200 mg every 3 weeks (*n* = 56) in first-line therapy (tumor tissue PD-L1 ≥50%) or with nivolumab (Nivo cohort, NC) at 3 mg/kg or 240 mg every two weeks (*n* = 126) in second or further lines, until disease progression, unacceptable toxicity, patient refusal, or death occurred. All patients underwent a CT-SCAN every 6–8 weeks and response assessment was performed in accordance with the response evaluation criteria in solid tumors (RECIST). For each patient included in the study, a peripheral blood sample was collected at the baseline (prior to any ICI therapy) and at the first CT-SCAN evaluation for the Pembro cohort only. The sample size was estimated from a subset (89/126) of NC patients using data from two circulating markers associated with OS [19], assuming a statistical power of 0.80, a two-tailed type I error of 0.05, an OS probability of 0.75 and a predicted withdrawal rate of 0.05, requiring 87 patients for each group.

### 2.2. Tumor PD-L1 Assessment

PD-L1 tumor expression was evaluated by immunohistochemistry on formalin-fixed paraffin-embedded tissue (>100 tumor cells) from the PC using the clone SP263 on the BenchMark ULTRA automatic slide-staining system (Ventana Medical System, Tucson, AZ, USA). PD-L1 expression was evaluated in the tumor cells according to the Tumor Proportion Score.

### 2.3. Soluble Biomarkers Evaluation

Plasma levels of sPD-L1, sB7-H3 and sB7-H4 were detected by ELISA using specific commercial kits: Human PD-L1 ELISA Kit [28-8] (ab277712) (Abcam, Cambridge, UK), Human B7-H3/CD276 ELISA (RayBiotech Life, Inc., Norcross GA, USA) and Human B7-H4 ELISA Kit (ab233633; Abcam). In each plate, 50 µL (PD-L1 and B7-H4) or 100 µL (B7-H3) of the plasma samples and standards were run in duplicate and analyzed according to the manufacturer’s instructions. The optical density was read at 450 nm using the Optic Ivyman system spectrophotometer (Biotech Madrid, España). sPD-L1 and sB7-H4 concentrations (pg/mL) were obtained by linearly interpolating the mean absorbance values subtracted by the blank control against the standard curve. The levels of sB7-H3 (ng/mL) were obtained by a double logarithmic (log-log) scale. In addition, for each patient, the interferon-gamma (IFNG) level was also assessed using the simple Plex Human IFN-gamma (3rd Gen) cartridge by Ella automated microfluidic platform (ProteinSimple, Bio-Techne, Minneapolis, MN, USA) using 50 µL of the diluted plasma samples (1:2 with the diluent). IFNG concentrations (pg/mL) were obtained using the manufacturer-calibrated standard curve and Ella software.

### 2.4. EV Isolation and Characterization

EVs were isolated from 500 µL of the plasma using qEV Original/70 nm (Izon, Christchurch, New Zeland) size-exclusion chromatography (SEC) columns, according to the manufacturer’s instructions. The size and concentration of collected EVs were analyzed by Nanoparticle Tracking Analysis (NTA) (NanoSight LM10, Malvern Instruments Ltd., Malvern, UK). The EVs were checked by transmission electron microscopy (TEM), as previously described [20]. EV protein expression was analyzed by western blot, as previously described [20], using an anti-flotillin-1 (1:10,000 dilution, ab41927, Abcam), anti-PD-L1 (1:1000 dilution, 13684, Cell Signaling, Danvers, MA, USA) and anti-B7-H3 antibody (1:1250 dilution, MAB1027-100, R&D Systems Biotechne, Minneapolis, MN, USA). Anti-rabbit (ECL-antirabbit IgG, NA934V, Amersham, dilution: 1:2500) or anti-mouse (ECL-anti mouse IgG, NA931V, Amersham, 1:2000 dilution) secondary antibodies were used.

### 2.5. Flow Cytometry Characterization of EVs

EV characterization by flow cytometry was performed as previously described [21]. Each EV preparation was stained with 1 µM of CFDA-SE (Vybrant™ CFDA SE Cell Tracer Kit, Thermo Fisher Scientific, Waltham, MA, USA) at 4 °C and room temperature (RT). The expression of the markers CD9 (APC Mouse Anti-Human CD9, Clone HI9a, 312108; BioLegend, San Diego, CA, USA), CD63 (PE-Cy7 Mouse Anti-Human CD63, Clone H5C6, 561982; BD Biosciences, San Jose, CA, USA), CD81 (BV421 Mouse Anti-Human CD81, Clone JS-81, 740079; BD Biosciences), PD-L1 (Rb mAb to PDL1, Clone 28-8 (APC), ab206967; abcam), B7-H3 (BB700 Mouse anti-human CD276, Clone 7-517, 745828; BD Biosciences) and B7-H4 (PE-CF594 Mouse anti-human B7-H4, Clone MIH43, 562785; BD Biosciences) was evaluated within the CFDA-SE-positive events and compared to the corresponding isotype controls using BD FACSAria II (BD Biosciences).

### 2.6. Multiplex EV Surface Marker Analysis

An analysis of surface antigen expression on the EVs was performed using the human MACSPlex Exosome kit (Miltenyi Biotec, Bergisch-Gladbach, Germany) following the manufacturer’s instructions, with a few modifications. Briefly, 7 µg of EV protein, measured by BCA Protein Assay Kit (Thermo Fisher Scientific) or PBS (blank control) was diluted in 120 µL of the MacsPlex buffer. The EVs were incubated in 15 μL capture beads overnight at 4 °C under gentle agitation and protection from light. The EV-bead complexes were washed using 1 mL MACSPlex buffer and centrifuged at 3000× *g* for 5 min. Detection antibody mixture was added to the beads and samples were incubated for 1 h, RT. In order to evaluate the origin of either PD-L1-positive or B7-H3-positive EVs, in parallel PD-L1 (Rb mAb to PDL1, Clone 28-8 (APC), ab206967, Abcam) was added to the EV–bead complexes instead of the detection antibody mixture. B7-H3 (BB700 Mouse Anti-Human CD276, Clone 7-517, 745828, BD Biosciences) was added, in additional tubes, to the EV–bead complexes together with the detection antibody mixture. The samples were washed and analyzed on BD FACSAria II (BD Biosciences). The background values of the PBS and isotype controls (REA or mouse IgG) were subtracted from each of the sample PE median fluorescence intensity values (MFI).

### 2.7. Statistical Analysis

Descriptive statistics was applied to report the characteristics of patients and diseases in the two clinical cohorts separately. Categorical variables (gender, smoking status, Eastern Cooperative Oncology Group Performance Status (ECOG-PS), tumor histology, tumor stage, line of therapy and cycles of therapy) were expressed as absolute numbers and relative frequencies (percentages), while age at diagnosis was summarized using the median and min–max range. Distributions of biomarker levels were described by means of the median, interquartile (P25–P75) and min–max ranges. An analysis of contingency tables and the Kaplan–Meier method were applied to inspect the prognostic role of the investigated biomarkers in response and survival outcomes. Indexes of association between biomarker levels and main clinical outcomes, adjusted to account for confounding effects attributable to imbalances in baseline patient/disease characteristics, were estimated using multivariable regression analyses. In particular, two distinct approaches were considered, namely the modified Poisson method [22] for tumor response data and the Cox method for PFS/OS data; the non-response rate ratio (RR) and progression/mortality HR were computed as indexes of relative risk along with corresponding 95% confidence limits (95% CL). In all the multivariable analyses all biomarkers entered the regression equation simultaneously (joint effect), and p-values derived from the likelihood ratio test were also calculated. All data analyses were performed using Stata software (StataCorp. Stata: Release 17, Statistical Software., College Station, TX, USA, 2021).

## 3. Results

### 3.1. Clinicopathologic Features of Patient Cohorts

Patient characteristics of the two cohorts are summarized in Table 1 and fully reported in Appendix A. In both cohorts, males were more represented than females and the median age was 70 years. At the time of starting therapy, patients had ECOG-PS = 1 in most cases (PC: 48%; NC: 65%) and the most frequent tumor type was adenocarcinoma (PC: 52%; NC: 71%). The median follow-up times were 12.7 months (0.27–43.4) in the PC and 8.7 months (0.53–70.9) in the NC. At the end of the follow-up period, 64.3% (36/56) of PC patients and 84.9% (107/126) of NC patients had died, with a median OS of 15.5 months (95% CL = 10.4–22.8) and 8.6 months (95% CL = 5.5–12.1), respectively. In addition, 89.3% (50/56) of the PC patients and 91.3% (115/126) of the NC patients had relapsed according to the RECIST criteria showing, respectively, a median PFS of 6.5 months (95% CL = 2.8–7.9) and 2.0 months (95% CL = 1.7–3.8).

### 3.2. Soluble Molecules Are Prognostic Markers in Patients Treated with Pembrolizumab

All baseline PC plasma samples were successfully processed while three plasma NC samples failed the tests (one sB7-H4 and two sB7-H3) (Appendix A). Generally, no remarkable difference in median levels was detected in both cohorts, although a slightly higher median concentration of sB7-H4 was measured in the PC than in the NC (63.9 pg/mL vs. 51.4 pg/mL) (Table 2).

In addition, we did not observe any correlation (Pearson coefficient = −0.046; *p*-value = 0.751) between sPD-L1 concentrations and corresponding tumor tissue expression in the PC. The prognostic role of the circulating biomarkers was estimated through the multivariable Cox regression analysis (Figure 1a). Figure 2 and Table 3 show the results of the joint effect of all the biomarkers on OS and PFS based on the median values of the soluble markers.

Overall, higher sPD-L1 levels (>24.2 pg/mL) were linked to worse survival outcomes (OS: HR = 1.77, 95% CL = 0.72–4.37; PFS: HR = 2.16, 95% CL = 1.08–4.33) in the PC patients, while both sB7-H biomarkers showed an inverse correlation (Figure 2; Appendix A). In particular, levels of sB7-H3 higher than 0.5 ng/mL were associated with better survival outcomes (OS: HR = 0.33, 95% CL = 0.14–0.78; PFS: HR = 0.32, 95% CL = 0.17–0.64), and a similar trend was also observed with sB7-H4 levels higher than 63.9 pg/mL (OS: HR = 0.42, 95% CL = 0.19–0.94; PFS: HR = 0.32, 95% CL = 0.16–0.64).

In the NC patients, only sB7-H3 was found to be associated with survival (Appendix A; Appendix A). Specifically, a decrease in mortality and progression rates of about 46% (HR = 0.54, 95% CL = 0.34–0.86) and 31% (HR = 0.69, 95% CL = 0.46–1.03), respectively, were estimated in patients with sB7-H3 levels higher than 0.4 ng/mL.

As far as IFNG was concerned, only the PC patients with higher levels (>1.0 pg/mL) had a mortality risk, which was about three times higher (HR = 2.95, 95% CL = 1.18–7.36) than that of patients with lower levels (Appendix A), resulting in significantly reduced OS.

In addition, the association between soluble markers and RECIST-based disease response to ICI therapy was also investigated by comparing patients with a disease control response (complete/partial response, and stable disease) to non-responders (NRs, patients with progression and early death), using multivariable Poisson regression. Non-response rates were found to be generally independent from all soluble markers in both cohorts, although a non-statistically significant association with higher values of sB7-H3 (RR = 0.51; 95% CL = 0.23–1.12) and of sB7-H4 (RR = 0.47; 95% CL = 0.25–0.86) was detected in the PC (Appendix A). Figure 3 summarizes the relative risks based on marker expression; notably, similar risk patterns in the three diverse outcomes were found only in the PC for sPD-L1, in which higher levels (sPD-L1 > 24.2 pg/mL) seemed to correlate to a higher risk (RR/HR > 1.0). Conversely, higher levels of both forms of sB7-H (sB7-H3 > 0.5 mg/mL; sB7-H4 > 63.9 pg/mL) appeared to play a protective role (RR/HR < 1.0).

### 3.3. PD-L1 and B7-H3 Are Associated to Plasma EVs

To assess whether soluble markers were derived from a free-cleaved form or were associated to EVs [23], we isolated and characterized plasma EVs from a subgroup of 9 out of 56 patients from the PC cohort (Appendix A). TEM and flowcytometry confirmed the presence of a heterogeneous EV population enriched in small-size EVs (Appendix A). Both PD-L1 and B7-H3 were detected in EVs derived from all analyzed patients (PD-L1: 87.0 ± 15.8; B7-H3: 57.2 ± 28.3), indicating that these two markers can be secreted through EVs (Appendix A), although we did not find any correlations (Bonferroni-adjusted *p*-value > 0.1). On the contrary, despite the high plasma levels of sB7-H4, the percentage of EVs expressing B7-H4 was relatively low (median: 10.7), suggesting that the circulating extracellular form of this marker might have mainly been derived from a proteolytic cleavage of the native protein (Appendix A).

### 3.4. EV-Associated PD-L1 and B7-H3 Predict the Response to ICI

To elucidate the role of EV-associated B7-H3 and PD-L1 in the response to ICI (Figure 1b), we focused on treatment-naïve patients (PC), thus avoiding any bias caused by previous treatments. EV characterization was performed on 20 PC patients selected based on therapy response (RECIST): (i) 11/20 NR (10 showing progression of disease and 1 with an early death); (ii) 9/20 responding patients (CR/PR: 2 complete response and 7 partial response) (Appendix A). For each patient, two time points were considered: the baseline (T0) and the first CT-scan assessment after pembrolizumab (T1). No significant differences were detected in terms of EV size among the samples (Appendix A), whereas EV concentration significantly increased in the NR patients at T1 compared to T0 (*p* < 0.01) (Appendix A). The tetraspanin family members (CD9, CD63 and CD81) were expressed in all samples but at different levels; CD81 and CD9 were the most expressed antigens with a mean of 96.6 ± 2 and 74.1 ± 11.6, respectively, while CD63 was generally less expressed (26.9 ± 13) (Appendix A). Interestingly, both PD-L1+ and B7-H3+ EVs were differentially expressed among the NR and CR/PR patients (Figure 4a–d). A significant increase in PD-L1+ EVs was detected in the NR patients when compared with the CR/PR patients at T1 (*p* < 0.0001) (Figure 4a,b). In addition, a relevant decrease of PD-L1+ EVs within the cohort of CR/PR patients was observed between T0 and T1 (*p* < 0.0001) (Figure 4b).

Despite the fact that B7-H3+ EV levels were generally higher in the NR patients than the CR/PR patients, we observed an increase in patients with CR/PR from T0 to T1 (Figure 4c). These findings were confirmed by the multivariable analysis, in which the potential effect of change in PD-L1 and B7-H3 EVs was expressed as a difference between measurements evaluated at T0 and T1 (Δ = T1 − T0). In particular, while an increasing trend in PD-L1+ EVs was found to be associated with a higher non-response rate (RR = 5.67; 95% CL = 1.80 − 17.9), a similar but inverse effect was observed in B7-H3+ EVs, which showed a reduction of about 70% (RR = 0.29; 95% CL = 0.11 − 0.81) (Appendix A).

### 3.5. PD-L1- and B7-H3-EVs Show Different Cell Origins

To assess the origin of either PD-L1- or B7-H3-expressing EVs from the PC patients, a MACSPlex multiplex assay was performed. We focused on 6 out 20 patients selected according to the RECIST response at both T0 and T1. Nine proteins (HLA-ABC, CD42a, CD40, CD62P, CD29, SSEA-4, CD41b, CD31 and CD69) were identified in PD-L1-expressing EVs after normalizing their expression against the CD9/CD63/CD81 mean fluorescence intensity (MFI) (Figure 5a).

Among the expressed proteins, platelet-associated markers (CD31, CD41b, CD42 and CD62P) were the most enriched. In particular, the expression of the platelet activation marker (CD62P) was generally enriched in the EVs from the NR patients (Figure 5b). In addition, PD-L1+ EVs expressed some tumor markers such as CD29 (integrin subunit beta 1) and stage-specific embryonic antigen-4 (SSEA-4). The latter one was mainly expressed in the NR rather than CR/PR patients (Figure 5b).

When the multiplex analysis was assessed for B7-H3+ EVs, two EV subpopulations with different B7-H3 fluorescent intensities were detected (Appendix A). The first one, expressing B7-H3 with a high fluorescence intensity (B7-H3^Bright^), showed an inverse trend in the NR and CR/PR patients between T0 and T1, specifically a decrease in the NR patients and an increase in the CR/PR patients (Figure 5c, Appendix A). The second subpopulation, expressing B7-H3 with a low fluorescence intensity (B7-H3^Dim^), was noticed in a small percentage of patients (Figure 5f, Appendix A). The two subpopulations were also distinguished based on the diverse expression of tetraspanin family members, in which B7-H3^Bright^ EVs mainly expressed CD9 (Figure 5d), whereas CD81 was the dominant tetraspanin characterizing B7-H3^Dim^ EVs (Figure 5g). Moreover, different profile proteins were distinguished in the two subpopulations: seven proteins (HLA-ABC, CD42a, CD40, CD62P, CD29, CD41b and CD31) were detected in B7-H3^Bright^ EVs after normalization against CD9 MFI (Figure 5e), and four proteins (HLA-DR, CD3, CD56 and CD45) were detected in B7-H3^Dim^ EVs after normalization against CD81 MFI (Figure 5h). Overall, the protein profile of B7-H3^Bright^ EVs was quite similar the one characterizing PD-L1-expressing EVs, although they lacked SSEA-4 and CD69 (Figure 5e). In addition, the expression of CD62P was mainly expressed in the NR patients rather than the CR/PR patients (Figure 5e). On the contrary, the antigenic profile of CD81-positive B7-H3^Dim^ EVs included immune-related proteins such as HLA-DR, the T cell marker CD3, the NK marker CD56 and the pan-leukocyte marker CD45 (Figure 5h).

## 4. Discussion

This study is the first to assess the prognostic role of soluble and EV-associated B7-H3 and B7-H4, together with the already known role of PD-L1, in a cohort of advanced NSCLC patients treated with ICIs in first and second-line settings. We found positive associations between sB7 markers and favorable outcomes, especially among patients in first-line treatment with pembrolizumab. Specifically, patients with elevated levels of sB7-H3 (>0.5 ng/mL) and sB7-H4 (>63.9 pg/mL) achieved longer OS and PFS. Notably, elevated levels of sB7-H3 were also associated with longer survival among the patients treated with nivolumab in second or subsequent lines. While both markers, when expressed on cancer cells, have been involved in tumor immune escape [24,25], some studies have also suggested that B7-H3 exerts an antitumor activity under certain conditions [26,27]. Similarly, controversial roles have also been described for the soluble forms. In particular, highly circulating levels of both markers have been linked to progressive disease in cancer patients [28,29,30]; on the contrary, high concentrations of sB7-H4 were linked to an enhancement of the immune response to autoimmune disease [31,32]. Azuma and colleagues reported that sB7-H4 might directly interact with the cell protein, blocking its immune inhibitory functions and enhancing the T-cell-mediated immune responses [31]. In line with these data, we also showed that the sB7-H4 form was mainly derived from a proteolytic cleavage of the native protein.

Regarding the soluble B7-H3 form, to date, no data on its involvement in the response to ICI have been reported. Interestingly, we noticed that the circulating form of B7-H3 was mainly associated with the EV surface, and that the increased percentage of B7-H3+ EVs was linked to a significantly reduced risk (70%) of progression after ICI treatment. The different B7-H3 densities also distinguished two EV subpopulations (B7-H3^Dim^ and B7-H3^Bright^). B7-H3^Dim^ EVs were mainly characterized by the expression of tetraspanin CD81 and by immune-related markers (i.e., CD56, CD3) whereas B7-H3^Bright^ EVs expressed tetraspanin CD9 and an antigenic profile quite similar to the one characterizing PD-L1-EVs, with the exception of the tumor marker SSEA-4 [33] and CD69. These data suggest the distinct origins of the EVs: the first one is mainly derived from NK and lymphocytes and the second one is more linked to platelets. In line with the first hypothesis, cell-surface B7-H3 has been identified in activated T cells [34] and infiltrating NK cells [35,36]. In the second population, although they were positive for platelet markers, the activation antigen (i.e., CD62P) was minimally expressed among patients showing a response (CR/PR) compared with non-responding patients. To date, growing evidence shows that platelet activation by cancer cells is the keystone toward a tumor-promoting phenotype [37,38]. Therefore, the reduction in platelet activation markers in B7-H3^Bright^ EVs derived from responders together with the inverse pattern of these vesicles among the patients at the first evaluation response (increased in responders and decreased in those who progress), support the hypothesis of B7-H3-EVs having a potential role in immune modulation.

We also found that elevated pre-treatment sPD-L1 levels (>24.2 pg/mL) were linked to worse survival in the PC, as already described in previous studies [14,39,40]. Notably, EVs isolated from the plasma of the PC patients expressed PD-L1, and their number significantly decreased at T1 in the responding patients. On the contrary, PD-L1+ EV increase was associated with up to a six-fold higher risk of progression. This finding is consistent with that of a previous study in which PD-L1+ EVs were increased among patients with advanced NSCLC who were not likely to benefit from ICIs [18]. An in-depth characterization of EV origins showed that PD-L1+ EVs mainly expressed the tumor antigen SSEA-4 and markers associated with resting (CD41b, CD42a) or activated (CD62P) platelets. The positivity of activated platelets was not surprising as platelets represent the major source of circulating EVs [41,42]. Following their interaction with cancer cells, the platelets can ingest PD-L1 and, in turn, present it on their surface [43]. In this regard, Hinterleitner and colleagues reported that NSCLC tumor cells themselves would transfer PD-L1 to platelets and that PD-L1 from platelets with high CD62P levels can also distinguish a subgroup of patients with shorter survival following ICI treatment [43]. All previous data have led to the hypothesis that PD-L1+ EVs might be involved in a mechanism of resistance to anti-PD-1 triggered by cancer cells, which directly or indirectly, through activated platelets, would release EVs expressing PD-L1. The latter might in turn interfere with anti-PD-1, leading to T-cell exhaustion.

In addition, elevated pre-treatment IFNG levels (>1.0 pg/mL) were also correlated to a reduction in OS. This cytokine is generally considered an antitumor immune factor, although debatable effects have also been reported [44,45]. Indeed, the persistent duration of IFNG signaling in tumor cells has been reported to regulate the resistance to ICIs by the activation of multiple inhibitory pathways [46]. In addition, other negative regulators of T-cell function, such as cytotoxic T-lymphocyte-associated antigen 4 (CTLA-4) are also induced by IFNG [47]. However, since PD-L1 tumor expression is mainly modulated by IFNG [48,49], and since in the present study the higher IFNG-induced risk of death was exclusively observed in the PC cohort with high tumor PD-L1 expression, we cannot rule out that the value of this might be biased.

Despite the advances made in our research, in terms of new non-invasive prognostic markers and the potential role of EVs in the immune response, our results should be interpreted in light of some reflections. In the first place, the need to split the cohorts into two further ones might have reduced the power of some statistical tests, specifically in the Pembro cohort (56 vs. the required 87). Nonetheless, we obtained in the Pembro cohort the most remarkable and meaningful results of both the PFS and OS analyses. This study, conducted in a single center, also lacked a validation cohort to test the prognostic effect of the single marker against. In addition, we did not observe remarkable associations of the three markers with patients’ prognosis in the NC, but the trends described in the PC cohort were generally preserved. However, it needs to be considered that the patients in NC experienced a high early mortality rate compared to patients in current real-life settings; this occurrence can be explained based on the fact that these patients had been recruited by the Expanded Access Program, allowing the inclusion of patients with negative prognostic characteristics such as brain metastases and poor PS, as well as heavily pre-treated patients. Finally, the EV characterization, albeit in-depth, was performed on a limited subset of patients. Nevertheless, our results on soluble and EV-associated forms provide the proof of concept of the potential connections of PD-L1 and B7-H3 expressing EVs to the ICI response.

## 5. Conclusions

To our knowledge, this is the first study that has assessed the prognostic role of plasmatic B7-H3 and B7-H4 in NSCLC patients treated with ICIs in first- and further-line settings. Our data highlight that circulating forms of sPD-L1, as well as sB7-H3 and sB7-H4, are emerging as non-invasive predictors of survival in ICI-treated metastatic NSCLC patients, acting independently of their tumor tissue expression. We also showed that dynamic changes in the expression of EV markers in the plasma of patients during therapy might be correlated with a different response to ICIs. Notably, an increase in PD-L1+ EVs was mainly related to disease progression, whereas an increase in B7-H3+ EVs appeared to be associated with ICI response. These data were also supported by the distinct protein profiles of the EV markers. The enrichment of tumor markers and activated platelets by PD-L1+ EVs, especially in the patients experiencing disease progression, suggests that EVs may participate in the escape from anticancer responses by competing with membrane PD-1 sites on T-cells, providing a barrier to protect tumors. On the contrary, the expression of immune antigens or resting platelet markers in B7-H3+ EVs and their increase during the first evaluation in responding patients leads to the speculation of their direct involvement in enhancing anti-tumor immunity. Our results, if further validated, may help set up multi-marker panels in the pursuit of developing a tailored therapy. In addition, the in vivo confirmation of the role of EVs in the response to ICIs might, in the near future, create new avenues for the development of EV-based therapeutic strategies to potentiate the immune system against tumor cells. 

## Figures and Tables

**Figure 1 cells-12-00832-f001:**
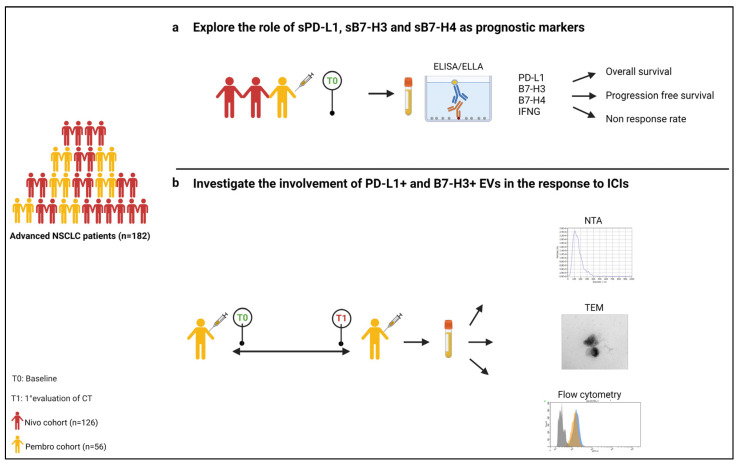
Study design: (**a**) explore the role of sPD-L1, sB7-H3 and sB7-H4 as prognostic markers in patients with advanced NSCLC treated with nivolumab or pembrolizumab; (**b**) assess the involvement of EVs expressing PD-L1 and B7-H3 in the mechanisms of the response to ICIs.

**Figure 2 cells-12-00832-f002:**
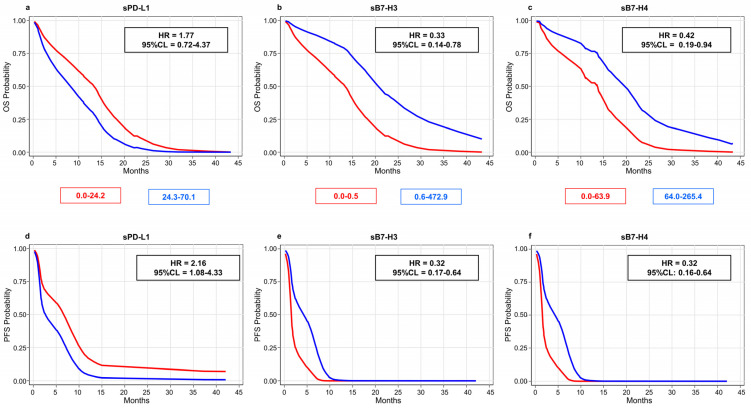
Overall (**a**–**c**) and progression-free (**d**–**f**) survival probabilities estimated through multivariable Cox regression analysis in 56 patients of Pembro cohort stratified according to biomarker median values: sPD-L1 = 24.2 pg/mL (**a**,**d**); sB7-H3 = 0.5 ng/mL (**b**,**e**); sB7-H4 = 63.9 pg/mL (**c**,**f**). Legend—HR: hazard rate ratio adjusted for gender, age, cycles of therapy, ECOG-PS and histotype; 95% CL: 95% confidence limits for HR.

**Figure 3 cells-12-00832-f003:**
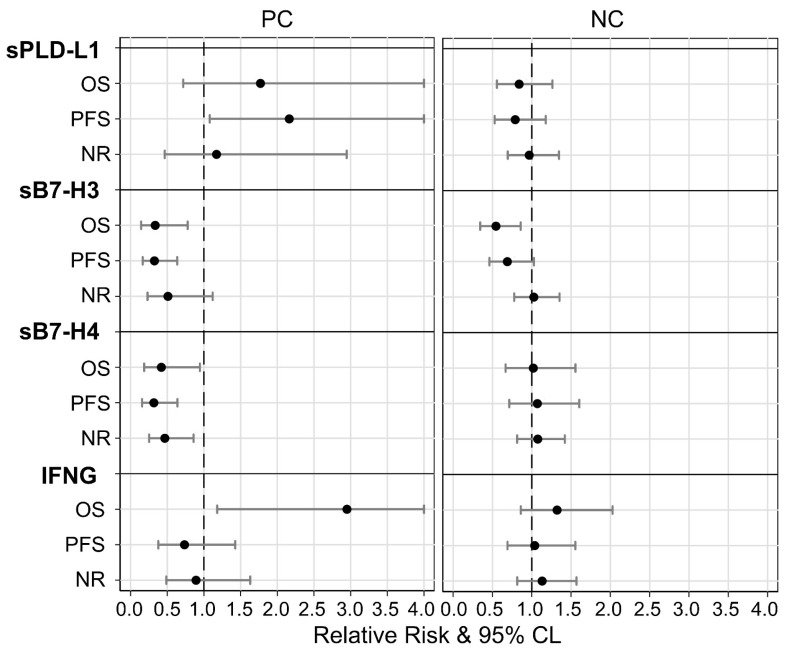
Comparisons between prognostic effects of study biomarkers in Pembro (PC) and Nivo (NC) cohorts on mortality (OS), progression (PFS) and non-response (NR) rates estimated through multivariable modified Poisson (NR) and Cox (OS and PFS) regressions. Legend—relative risk: (black points) ratio between adverse outcome rates for patients with higher vs. lower biomarker levels; 95% CL: (gray whiskers) 95% confidence limits for relative risk. Note—all relative risks were adjusted for gender, age, cycles, ECOG and histotype; relative risk = 1: (vertical dashed line) equal rates in both biomarker categories; relative risk >1: greater rates in higher categories; relative risk <1: lower rates in higher categories.

**Figure 4 cells-12-00832-f004:**
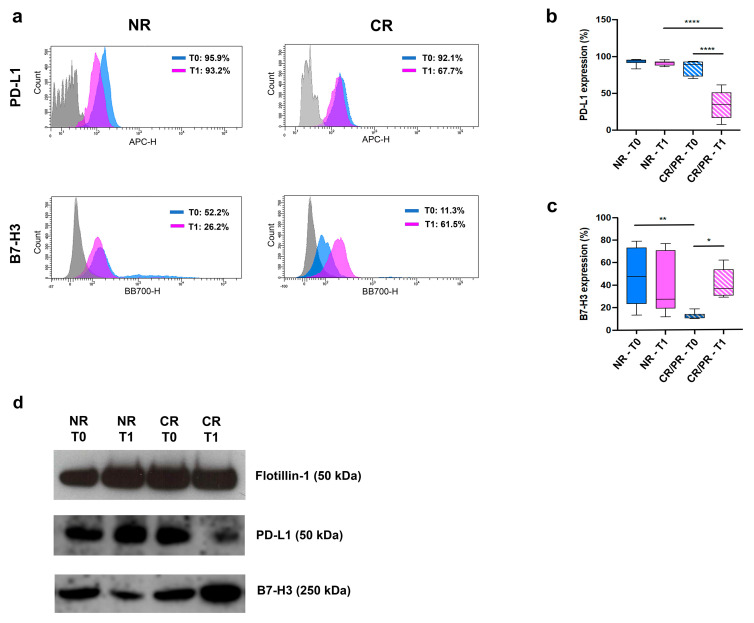
(**a**) Representative flow cytometry analysis of EVs derived from plasma of NR (#146) and CR (#174) patients, at both T0 and T1. Areas under blue and pink lines indicate EVs reacting with PD-L1 (upper panels) or B7-H3 (bottom panels) at T0 and T1, respectively. Areas under the grey lines indicate the interactions of vesicles with corresponding non-reactive immunoglobulin of the same isotype. (**b**,**c**) Histograms representing the percentage of PD-L1- (**b**) or B7-H3-positive EVs (**c**) derived from NR and CR/PR patients, at both T0 and T1. *: *p* < 0.05, **: *p* < 0.01, ****: *p* < 0.0001 (one-way ANOVA). (**d**) Western blot analysis of Flotillin-1, PD-L1 and B7-H3 on EVs derived from plasma of NR (#146) and CR (#174) patients, at both T0 and T1.

**Figure 5 cells-12-00832-f005:**
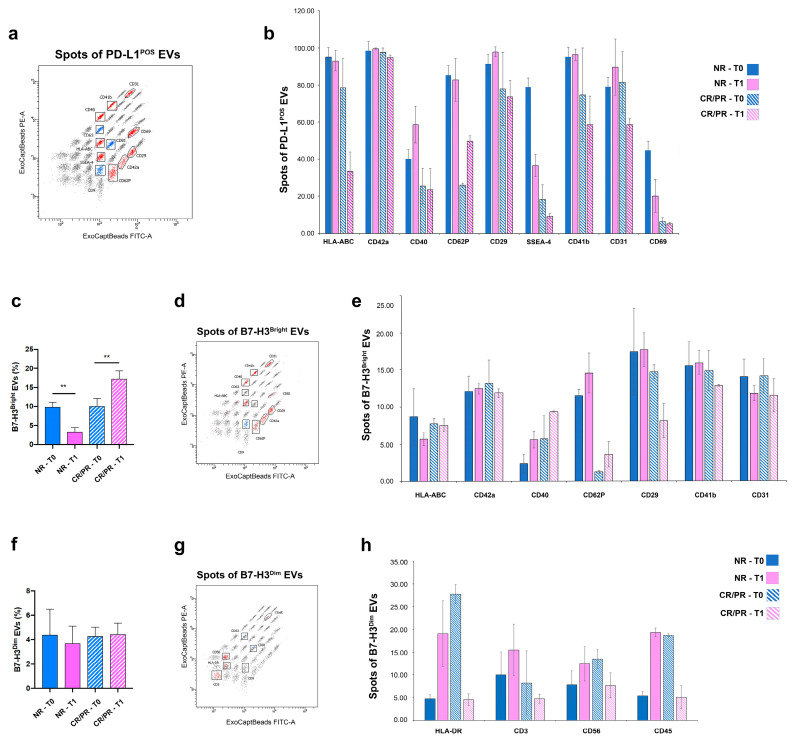
EV surface proteins profiled by multiplex bead-based flow cytometry assay. Captured EVs were counterstained with APC-labelled detection antibodies (mixture of anti-CD9, -CD63 and -CD81 antibodies). (**a**,**d**,**g**) Dot charts of beads corresponding to each of the 37 proteins. Red dots indicate: (**a**) PD-L1-expressing EVs identified above detection threshold in any of the patients, after normalizing their expression against CD9/CD63/CD81 MFI; (**d**) B7-H3^Bright^ EVs identified above detection threshold after normalizing their expression against CD9 MFI; (**g**) B7-H3^Dim^ EVs identified above detection threshold after normalizing their expression against CD81 MFI. Blue dots correspond to tetraspanin marker whereas black dots correspond to negative markers. (**b**,**e**,**h**) Histograms represent: (**b**) 9 out of 37 proteins that were found to be positive in PD-L1-expressing EVs isolated from any of the patients; (**e**) 7 out of 37 proteins that were found to be positive in B7-H3^Bright^ EVs isolated from any of the patients; (**h**) 4 out of 37 proteins that were found to be positive in B7-H3^Dim^ EVs isolated from any patients. (**c**,**f**) Histograms represent: percentage of B7-H3^Bright^ (**c**) B7-H3^Dim^ (**f**) and EVs isolated from NR and CR/PR patients at both T0 and T1. Error bars in graphs represent mean ± SD. ** *p* < 0.01 (Ordinary one-way ANOVA).

**Table 1 cells-12-00832-t001:** List of the most relevant clinicopathological characteristics of patients treated in first-line therapy (Pembro cohort, PC) or in second/further lines of treatment (Nivo cohort, NC).

Characteristics	PC	NC
**Median age at start of therapy (Range)**	70.1 (50.5–88.8)	70.1 (44.2–87.6)
**Gender**	**N (%)**	**N (%)**
Male	43 (76.8)	91 (72.2)
Female	13 (23.2)	35 (27.8)
**Smoking habit**	**N (%)**	**N (%)**
Never	2 (3.6)	10 (7.9)
Former	31 (55.4)	70 (55.6)
Current	21 (37.5)	44 (34.9)
Missing	2 (3.6)	2 (1.6)
**ECOG-PS**	**N (%)**	**N (%)**
0	19 (33.9)	29 (23.0)
1	27 (48.2)	82 (65.1)
2	10 (17.9)	14 (11.1)
3	0 (0.0)	1 (0.8)
**Histotype**	**N (%)**	**N (%)**
Adenocarcinoma	29 (51.8)	90 (71.4)
Squamous cell carcinoma	14 (25.0)	30 (23.8)
Other	13 (23.2)	6 (4.8)
**Stage**	**N (%)**	**N (%)**
IIIB	0 (0.0)	6 (4.8)
IV	56 (100.0)	120 (95.2)
**Therapy line**	**N (%)**	**N (%)**
1st	56 (100)	0 (0.0)
2nd	0 (0.0)	76 (60.3)
3rd	0 (0.0)	29 (23.0)
Other	0 (0.0)	21 (16.7)
**Cycles of therapy received**	**N (%)**	**N (%)**
1–8	27 (48.2)	69 (54.8)
9–123	29 (51.8)	56 (44.4)
Missing	0 (0.0)	1 (0.8)
**Total**	56 (100.0)	126 (100.0)

**Table 2 cells-12-00832-t002:** Statistical parameters of biomarker distributions in Pembro and Nivo cohorts at baseline.

Cohort	Biomarkers (Units)	N	Median	P25	P75	Range
**Pembro**	**sPD-L1** (pg/mL)	56	24.2	13.9	32.7	0.0–70.1
	**sB7-H3** (ng/mL)	56	0.5	0.1	1.9	0.0–472.9
	**sB7-H4** (pg/mL)	56	63.9	13.3	141.6	0.0–265.4
	**IFNG** (pg/mL)	56	1	0.7	1.5	0.2–4.1
**Nivo**	**sPD-L1** (pg/mL)	126	24.7	17.5	36.7	0.0–94.2
	**sB7-H3** (ng/mL)	124	0.4	0	1.2	0.0–33.6
	**sB7-H4** (pg/mL)	125	51.4	27.9	89.2	0.0–738.1
	**IFNG** (pg/mL)	126	1	0.7	1.9	0.4–16.1

Legend—N: absolute frequency; P25/P75: 25°/75° percentile; range: min–max values.

**Table 3 cells-12-00832-t003:** Joint effect of all biomarkers on RECIST-based progression and mortality rates in Pembro and Nivo cohorts estimated through the multivariable Cox regression analysis.

Cohort	Biomarker and Levels	PFS (RECIST)			OS		
		HR	95% CL	*p*-Value	HR	95% CL	*p*-Value
**PC**	**sPD-L1**			0.030 *			0.215
	0.0–24.2	1.00	(Ref.)		1.00	(Ref.)	
	24.3–70.1	2.16	1.08–4.33		1.77	0.72–4.37	
	**sB7-H3**			0.001 *			0.011 *
	0.0–0.5	1.00	(Ref.)		1.00	(Ref.)	
	0.6–472.9	0.32	0.17–0.64		0.33	0.14–0.78	
	**sB7-H4**			0.001 *			0.036 *
	0.0–63.9	1.00	(Ref.)		1.00	(Ref.)	
	64.0–265.4	0.32	0.16–0.64		0.42	0.19–0.94	
	**IFNG**			0.362			0.021 *
	0.2–1.0	1.00	(Ref.)		1.00	(Ref.)	
	1.1–4.1	0.73	0.38–1.43		2.95	1.18–7.36	
**NC**	**sPD-L1**			0.247			0.399
	0.0–24.7	1.00	(Ref.)		1.00	(Ref.)	
	24.8–94.2	0.79	0.53–1.18		0.84	0.56–1.26	
	**sB7-H3**			0.068			0.009 *
	0.0–0.4	1.00	(Ref.)		1.00	(Ref.)	
	0.5–33.6	0.69	0.46–1.03		0.54	0.34–0.86	
	**sB7-H4**			0.743			0.932
	0.0–51.4	1.00	(Ref.)		1.00	(Ref.)	
	51.5–738.1	1.07	0.71–1.60		1.02	0.67–1.56	
	**IFNG**			0.861			0.203
	0.4–1.0	1.00	(Ref.)		1.00	(Ref.)	
	1.1–16.1	1.04	0.69–1.55		1.32	0.86–2.03	

Legend—HR: hazard rate ratio adjusted for gender, age, cycles of therapy, ECOG-PS and histotype; 95% CL: 95% confidence limits for HR; *p*-value: probability level associated with the likelihood ratio test result; ref.: reference category (lower levels of biomarkers). * *p* < 0.05.

## Data Availability

Data supporting our findings are included in the Appendix A. The fcs files from the MACSPlex Exosome kit (Miltenyi) generated during the current study are available in FlowRepository (https://flowrepository.org/; ID: FR-FCM-Z5UQ; accessed on 24 November 2022).

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
