# Peer review of "Prognostic Role of Soluble and Extracellular Vesicle-Associated PD-L1, B7-H3 and B7-H4 in Non-Small Cell Lung Cancer Patients Treated with Immune Checkpoint Inhibitors"

_cells, 2023, doi:10.3390/cells12060832_

Round 1
Reviewer 1 Report
The manuscript is well written and only requires minor language corrections.
Presented data may have practical meaning and suggest a potential EV involvement in ICI response which should be explored further.
Author Response
Reviewer #1:
The manuscript is well written and only requires minor language corrections. Presented data may have practical meaning and suggest a potential EV involvement in ICI response which should be explored further.
Dear reviewer#1 thank you for your suggestions; hereby our reply to the revision:
We thank the reviewer for expressing a favorable opinion on our study. Indeed, our data, if confirmed in in vivo models, could, in the near future, open new avenues for the development of EV-based therapeutic strategies to enhance the immune system against tumor cells. As pointed out by the reviewer, we have also added this relevant point in the Conclusion (lines: 532-535).
Reviewer 2 Report
The article sheds more light on the possibility of finding new biomarkers for the efficacy of immunotherapy in NSCLC. The identification of the potential prognostic role of soluble B7-H3 and B7-H4 represents a new data, compared to the already known soluble PD-L1.
However, the work is lacking in the study design. In fact, the scientific questions to be answered are not well specified. Accordingly, they do not describe a method for calculating the appropriate sample size to study, based on statistical power and the expected effect size. A further consequence of this lack of study design can be found in the conclusions, in which there is no clear answer (positive or negative) to the initial questions. Moreover, the inclusion of two cohorts from different treatment lines with different drugs limits even more the power of these findings. Thus, the results described and the statistical analyses appear correct and interesting, but the study design and its structure need to be explained better.
Author Response
Reviewer #2
The article sheds more light on the possibility of finding new biomarkers for the efficacy of immunotherapy in NSCLC. The identification of the potential prognostic role of soluble B7-H3 and B7-H4 represents a new data, compared to the already known soluble PD-L1.
However, the work is lacking in the study design. In fact, the scientific questions to be answered are not well specified. Accordingly, they do not describe a method for calculating the appropriate sample size to study, based on statistical power and the expected effect size. A further consequence of this lack of study design can be found in the conclusions, in which there is no clear answer (positive or negative) to the initial questions. Moreover, the inclusion of two cohorts from different treatment lines with different drugs limits even more the power of these findings. Thus, the results described and the statistical analyses appear correct and interesting, but the study design and its structure need to be explained better.
Dear reviewer #2: We thank you for your helpful comments and apologize for the lack of a clear study design in our manuscript.
Our preclinical observational study aims to investigate two main points: 1) to explore the role of the soluble forms of two members of the B7-H family (i.e., B7-H3 and B7-H4), together with the already known programmed death ligand 1 (PD -L1), as prognostic markers in a cohort of patients with advanced NSCLC treated with anti-PD-1 in first- or second-line treatment; 2) to investigate the potential involvement of EV positive for these markers in the response to ICIs (Figure 1)
Figure 1. Study design: a) explore the role of sPD-L1, sB7-H3 and sB7-H4 as prognostic markers in patients with advanced NSCLC treated with nivolumab or pembrolizumab; b) assess the involvement of EVs expressing PD-L1 and B7-H3 in the mechanisms of response to ICI.
For the first study, we included a cohort of 182 consecutive advanced NSCLC patients receiving ICIs from May 2015 to May 2019 included in a mono-institutional translational research.
Before starting our study, we estimated the sample size using data already published including a subset of patients (89/126) belonging to the cohort treated with nivolumab (PMID: 31295929).
In order to have a plausible estimate of Hazard Ratio (HR) when comparing better versus worse prognostic categories, we exploited the results of two circulating biomarkers, namely Circulating Tumor Cells (CTCs) and circulating Free DNA (cfDNA), proved to be significantly associated with overall survival (OS).
In particular, in that study we were able to distinguish two biomarker categories (CTCs ≤ 2 and cfDNA ≤ 836.5) from which an average mortality rate (HR) of about 0.45 estimated when patients at a lower death risk were compared to those at higher risk. Accordingly, assuming a statistical power of 0.80, a two-tailed type I error of 0.05, an OS probability of 0.75 and a predicted withdrawal rate of 0.05, about 90 (87) patients were needed for analysis (Figure 2, green line).
Fig. 2. Sample size estimation according to statistical power and expected HR between patients with unfavorable and favorable soluble marker.
Despite the Nivo cohort was largely fulfilled (126 compared 87 needed patients), during the recruitment period we enrolled only 56 patients in the Pembro cohort, representing 64% (56/87) of the anticipated sample size. Definitely, this limitation amounted to a reduction in statistical power. Indeed, considering a two-tailed type I error of 0.05, HR = 0.42 and an observed death probability of 0.65 (both statistics estimated from our current data), we have obtained a power of about 0.65 for the present study (Figure 3, green line).
Fig. 3. Statistical power and expected HR estimation according to sample size.
However, even with this important limitation we obtained in Pembro cohort the most remarkable and meaningful results both in PFS and OS analyses compared to the well-sized Nivo cohort.
In the second part of the study, we focused on the involvement of EVs in therapy response, and compared to the previous study is observational only. In particular, we initially performed a pilot study on a very limited subset of patients (9/56) to verify the expression of three markers on EV surfaces. Patients were selected on the basis of different soluble levels of three markers, although we did not find significant correlations between soluble portions and EV expression (Bonferroni-adjusted p-value>0.1). Furthermore, the analysis also showed that despite high plasma levels of sB7-H4, the percentage of EVs expressing B7-H4 was relatively low (median: 10.7).
This result led to the hypothesis that the circulating extracellular form of B7-H4 could derive mainly from a proteolytic cleavage of the native protein. The previous findings are not surprising; indeed, some studies reported that the expression of immune-checkpoint inhibitors, such as PD-L1, by EVs was not correlated with the soluble levels or with the expression at tumor tissue level (PMID: 32002173).
Thus, since both PD-L1 and B7-H3 were expressed on the EV surface and current evidence demonstrates a direct role of EVs in the ICI response (PMID: 36484171; PMID: 35650597), we also investigated PD-L1 and B7-H3 EVs in our patient cohort. In particular, to try to answer this relevant question, we selected 20 among 56 naïve patients. We focused on this cohort to avoid any bias due to prior lines of therapy in the nivolumab cohort. Patients were selected based on response to therapy (RECIST): i) 11/20 non-responders (NR: 10 progressive disease and 1 early death); ii) 9/20 responders (CR/PR: 2 complete responses and 7 partial responses).
Interestingly, when we compared baseline and first evaluation, we observed higher EV concentration in NR patients at T1 compared to T0 (p < 0.01) (Figure S4c). Furthermore, both PD-L1+ and B7-H3+ EVs were differentially expressed between NR and CR/PR patients and between T0 and T1 (Figure 4a–d). In particular, a significant increase of PD-L1+ EVs was detected in NR compared with CR/PR patients at T1 (p<0.0001) (Figure 4a,b), whereas a decrease of PD-L1+ EVs within the cohort of CR/PR patients was observed between T0 and T1 (p<0.0001) (Figure 4b). On the contrary, the B7-H3+ EV levels increased in patients with CR/PR from T0 to T1 (Figure 4c). These findings were also confirmed in the multivariable analysis, where a potential effect of change in PD-L1 and B7-H3 EVs was expressed as difference between measurements evaluated at T0 and T1 (Δ =T1 – T0). In particular, while an increasing trend in PD-L1-EVs resulted to be associated with a higher non-response rate (RR= 5.67; 95%CI = 1.80-17.9), a similar but inverse effect was observed in B7-H3-EVs, that showed a reduction of about 70% (RR = 0.29; 95% CL = 0.11-0.81) (Table S8).
The further EV characterization showed an enrichment of tumor markers and activated platelets in PD-L1+ EVs, especially in patients experiencing disease progression, suggesting that EVs may participate in the escape from anticancer responses by competing with membrane PD-1 sites on T-cells, providing a barrier to protect tumors. On the contrary, the expression of immune antigens or resting platelet markers in B7-H3+EVs and their increase at the first evaluation in responding patients, leads to speculate their direct involvement in enhancing the anti-tumor immunity.
These dynamic modifications of EVs, in terms of frequency and antigenic profiles, allow us to verify our initial hypothesis on their potential involvement in the response to ICI.
We are aware that this second part of the study is purely observational, , although it provides a “Proof of Concept” of potential connections of EVs expressing PD-L1 and B7-H3 and ICI response; moreover, if our findings will be validated might, in the near future, open new avenues for the development of EV-based therapeutic strategies to potentiate the immune system against tumor cells.
All previous explications have been included in the new version of the manuscript together with the figure 1 summarizing the study design, as follows:
- In particular, high levels of these markers have been described as unfavorable factors in NSCLC [13], but to date there are no data in patients treated with ICIs (Lines: 81-83)
- EVs, acting as intercellular messengers by transferring protein and genetic materials, plays an active role in tumor-associated immune-cells communication [15], and in the immune response under ICIs [17,18]. To date, the involvement of EV-associated B7-H3 and B7-H4 in first- and second-line treated NSCLC patients is still unknown. The present study aims to: i) explore the role of sPD-L1, sB7-H3 and sB7-H4 as prognostic markers in patients with advanced NSCLC treated with nivolumab or pembrolizumab; ii) assess the involvement of EVs expressing PD-L1, B7-H3 and B7-H4 in the mechanisms of response to ICIs (Figure 1). (Lines: 86-93)
- The sample size was estimated in a subset (89/126) of NC patients using data from two circulating markers associated with OS [19], assuming a statistical power of 0.80, a two-tailed type I error of 0.05, an OS probability of 0.75 and a predicted withdrawal rate of 0.05, requiring 87 patients for each group. (lines: 122-125)
- The relationship prognostic role of circulating biomarkers was estimated through the multivariable Cox regression analysis (Figure 1a). Figure 2 and Table 3 show the results of the joint effect of all study biomarkers on OS and PFS based on the median values of soluble markers (Lines: 237-240)
- Both PD-L1 and B7-H3 were detected in EVs derived from all analyzed patients (PD-L1: 87.0 +/-15.8; B7-H3: 57.2 +/-28.3), indicating that these two markers can be secreted through EVs (Figure S3c), although we did not find any correlations (Bonferroni-adjusted p-value>0.1). (Lines: 335-338)
- To elucidate the role of EV-associated B7-H3 and PD-L1 in the ICI response (Figure 1b), we focused on treatment-naïve patients (PC), thus avoiding any bias due to previous treatments. (lines: 343-345).
- In first place, the need to split in two further cohorts might have reduced the power of some statistical tests specifically in the pembro cohort (56 vs. 87 required). Nonetheless, we obtained in Pembro cohort the most remarkable and meaningful results both in PFS and OS analyses (lines: 501-503).
- In addition, the in vivo confirmation of EV role in the ICI response might, in the near future, open new avenues for the development of EV-based therapeutic strategies to potentiate the immune system against tumor cells. (Lines: 532-535)

Reviewer 3 Report
The article "Prognostic role of soluble and extracellular vesicle-associated PD-L1, B7-H3 and B7-H4 in non-small cell lung cancer patients treated with immune checkpoint inhibitors" by Carlo Genova et al is focuses on the current and interesting topic of the prognostic significance of soluble and exosomal immune checkpoints in NSCLC patients receiving immunotherapy. The authors analyzed for the first time the association of soluble and exosomal forms of B7-H3 and B7-H4 with the response to pembrolizumab and nivolumab therapy. The authors also demonstrated that PD-L1 and B7-H3 were detected in EVs, while the percentage of EVs expressing B7-H4 was relatively low, suggesting that the circulating extracellular form of this marker might mainly derive from a proteolytic cleavage of the native protein. In general, the origin and functions of soluble forms of immune checkpoint proteins has not been fully determined to date and its study is extremely promising in the context of the development of non-invasive prognostic markers. The Introduction comprehensively describes the state of the art. Materials and Methods are well designed, to make the study reproducible. The results cover all necessary fields. The discussion is well thought out and supports the results, but some additional information could be added about the prognostic role of soluble forms of B7-H3 and B7-H4 in various types of tumors [e.g., https://doi.org/10.1007/s10517-021-05253-w; https://doi.org/10.1186/s12935-018-0614-z;https://doi.org/10.1186/s12935-018-0614-z; DOI:10.1371/journal.pone.0199719], etc. The Conclusions section reflects the main idea of the article.
Author Response
Reviewer 3
The article "Prognostic role of soluble and extracellular vesicle-associated PD-L1, B7-H3 and B7-H4 in non-small cell lung cancer patients treated with immune checkpoint inhibitors" by Carlo Genova et al is focuses on the current and interesting topic of the prognostic significance of soluble and exosomal immune checkpoints in NSCLC patients receiving immunotherapy. The authors analyzed for the first time the association of soluble and exosomal forms of B7-H3 and B7-H4 with the response to pembrolizumab and nivolumab therapy. The authors also demonstrated that PD-L1 and B7-H3 were detected in EVs, while the percentage of EVs expressing B7-H4 was relatively low, suggesting that the circulating extracellular form of this marker might mainly derive from a proteolytic cleavage of the native protein. In general, the origin and functions of soluble forms of immune checkpoint proteins has not been fully determined to date and its study is extremely promising in the context of the development of non-invasive prognostic markers. The Introduction comprehensively describes the state of the art. Materials and Methods are well designed, to make the study reproducible. The results cover all necessary fields. The discussion is well thought out and supports the results, but some additional information could be added about the prognostic role of soluble forms of B7-H3 and B7-H4 in various types of tumors [e.g., https://doi.org/10.1007/s10517-021-05253-w; https://doi.org/10.1186/s12935-018-0614-z;
https://doi.org/10.1186/s12935-018-0614-z; DOI:10.1371/journal.pone.0199719], etc.
The Conclusions section reflects the main idea of the article.
Dear reviewer #3: We thank the reviewer for expressing a favorable opinion on our study and for your helpful suggestions. Now, we have included these relevant studies on the role of both B7-H3 and B7-H4 markers and the disease progression in the Discussion as follows: “Similarly, controversial roles have also been described for the soluble forms. In particular, high circulating levels of both markers have been linked to progressive disease in cancer patients [29–31]; on the contrary high concentrations of sB7-H4 were linked to an enhancement of the immune response in the autoimmune disease [32,33].” (Lines: 440-444).
Round 2
Reviewer 2 Report
No more changes required